# Volumetric Carotid Flow Characteristics in Doppler Ultrasonography in Healthy Population Over 65 Years Old

**DOI:** 10.3390/jcm9051375

**Published:** 2020-05-07

**Authors:** Piotr Kaszczewski, Michal Elwertowski, Jerzy Leszczynski, Tomasz Ostrowski, Zbigniew Galazka

**Affiliations:** Department of General, Endocrine and Vascular Surgery, Medical University of Warsaw, Zwirki i Wigury 61 Street, 02-091 Warsaw, Poland; elwertowski.michal@gmail.com (M.E.); jleszcz1@gmail.com (J.L.); tostr@vp.pl (T.O.); zbigniew.galazka@wum.edu.pl (Z.G.)

**Keywords:** carotid flow volume, extracranial arteries, volumetric assessment

## Abstract

Background: Carotid flow velocity criteria are well established, with age being a factor influencing measurements. However, there are no volumetric standards for the flow in extracranial arteries. The aim of the study was related to volumetric flow assessment of extracranial arteries in a healthy population >65 years old. Methods: Doppler volumetric measurements of internal carotid (ICA), external carotid (ECA) and vertebral arteries (VA) were performed in 123 healthy volunteers >65 years old and compared with 56 healthy volunteers <65 years old. Results: The continuous decline in cerebral blood flow (CBF) volume was observed (*p* < 0.00001). Volumetric reference values were established in study groups: 1., 65–69 years: 898.5 ± 119.1; 2., 70–74 years: 838.5 ± 148.9; 3., 75–79 years: 805.1 ± 99.3; 4., >80 years: 685.7 ± 112.3 (mL/min). Significant differences were observed between groups: 1 and 3.4, as well as 3 and 4 (*p* = 0.0295, < 0.000001, 0.00446 respectively). CBF volume decreases gradually with age: 28–64 years—6.2 mL/year (*p* = 0.0019), 65–75 years—11.4 mL/year (*p* = 0.0121) and >75 years—14.3 mL/year (*p* = 0.0074). This is a consequence of flow volume decline in ICA (*p* = 0.00001) and to lesser extent ECA (*p* = 0.0011). The decrease of peak systolic (*p* = 0.002) and end diastolic (*p* = < 0.00001) velocities in ICA and peak systolic velocity in ECA (*p* = 0.0017) were observed. Conclusions: CBF decreases with ageing. Volumetric assessment of CBF may play an important additional role in diagnostics of patients with carotid stenosis. Doppler assessment of cerebral flow volume may create an interesting tool for identifying patients with diminished cerebrovascular reserve and higher risk of ischemic symptoms occurrence.

## 1. Introduction

Peripheral artery disease is estimated to affect about 20% of population beyond 60 years old and more than 50% over 85 years [1] and its prevalence is going to increase with the phenomenon of aging of the population [2]. Cardiovascular diseases are contemporarily considered one of the most serious health burdens, posing a first place among causes of death according to the WHO (Word Health Organization). In 2016, almost 18 million people died due to cardiovascular disorders, among them 85% because of myocardial infraction or stroke [3,4]. Stroke is estimated to affect annually about 15 million people worldwide. One third of these patients die while about 5 million remain permanently disabled [5]. It is regarded that about 85% of all strokes are of an ischemic type, while 15%, even up to one fifth, are the result of hemodynamically significant atherosclerotic lesions of the bifurcation of the common carotid artery and proximal part of internal carotid artery [6,7,8]. The quantitative assessment of the total cerebral blood flow previously required nuclear medicine methods, which availability is limited due to high costs and required equipment. The evaluation of total cerebral blood flow with ultrasound methods is featured with good accuracy, availability, and might be useful in monitoring various cerebrovascular disorders. While the blood flow velocity criteria in healthy patients as well as in the diagnosis of carotid stenosis are well defined, there are no reference values concerning blood flow volume. Limited data in the literature concerning the blood flow volume in carotid arteries are available, with only a few authors touching this problem, and research based on relatively small groups of patients [9,10,11,12]. The objective of this study covered assessment of the blood flow volume in internal carotid, external carotid and vertebral arteries in healthy volunteers over 65 years old in order to assess the normal hemodynamic parameters in this age group. The reference flow volume data might be potentially used in the diagnosis and monitoring the patients with various cerebrovascular disorders.

## 2. Subjects and Methods

In total, 123 healthy volunteers beyond 65 years old were included in the study (62 female; mean age 73.6 ± 6.6 years old, 61 male; mean age 72 ± 5.0 years old). In order to achieve homogenous stratification in terms of age, the whole study group was divided into cohorts in each 5-year period between 65 and 80 years of age, gender-disaggregated. Patients exceeding 80 years were gathered in one cohort. Maximum age in females was 96 years old and 87 years old in males. The group of 56 healthy volunteers aged 18–65 (34 female; mean age 51.2 ± 10.6 years, 22 male; mean age 49 ± 10.1 years) were included in the examination with the aim of comparison of flow parameters with the group over 65 years old. Characteristics of the study group is presented in the Table 1.

All patients lived an independent life and had no previous history of neurological and cerebrovascular disorders. Detailed inclusion and exclusion criteria are presented in Table 2. Informed consent was given by all study participants before the examination. The study was held with the approval of Medical University of Warsaw Bioethical Committee.

Before the Doppler ultrasonography (DUS) examination, blood pressure in each individual did not exceed 140 mmHg-systolic and 90 mmHg-diastolic, no tachycardia, or bradycardia was observed. In order to avoid interobserver variability, all measurements were performed by the same sonographer using the Canon Aplio i800 ultrasound scanner with Linear i11LX3 transducer. Each DUS began after 15 minutes of rest in supine position. During the examination, the patient lied supine with their head slightly elevated and turned to the contralateral side (by 10–20 degrees to the measurement of common carotid artery (CCA) and vertebral artery (VA), and about 30–40 degrees to internal carotid (ICA) and external carotid artery (ECA) measurements. The diameter of each vessel, defined as the shortest distance between internal layers of vessel walls, were measured 3 times with three different techniques: B-mode, SMI-Superb Micro-Vascular Imaging mode and hybrid: B-mode combined with SMI image. The average from 3 measurements was considered the vessel diameter. The cerebral blood flow was calculated as the sum of the flow volume in internal, external and vertebral arteries. The ICA flow measurements were conducted in their upper segments 3–4 cm above the ICA bulb to avoid false results due to changes in velocity in the vicinity of the carotid bulb. The external carotid arteries were evaluated above the origin of the upper thyroid artery to avoid false results due to physiological blood supply to the thyroid gland. Vertebral arteries were evaluated in possibly straight segments at V2 (V1 in the case of a tortuous course and technical difficulties). Flow volume in CCA was measured 4 cm below the carotid bifurcation. CCA flow was measured as a control—the measurement was regarded as accurate when CCA flow volume slightly exceeded the sum of ICA and ECA flow volume. All measurements were conducted three times and their average was considered a final result. Insonation angle was equal or lower than 60 degrees. The sample volume size was adjusted to cover between 0.5 and 2/3 of arterial lumen. The blood flow velocity wave forms were recorded—at least 5 waveforms with similar patterns were considered a proper flow pattern sample demanded to perform further measurements. The blood flow volumes were calculated using the ultrasound scanner semiautomatic program.

## 3. Statistics

Statistical analysis was performed with Statistica 13 (StatSoft Polska Sp. z.o.o., Krakow, Poland). The *t*-test, Mann–Whitney U test, analysis of variance (ANOVA) and regression analysis were performed. A *t*-test was applied when the normal distribution of data was stated. The Shapiro–Wilk test was performed as a test of normality: a data set with a p value of less than 0.05 rejects the null hypothesis that the data are from a normally distributed population. Consecutively, Levene’s test was used to assess the equality of variances—the *p*-value below 0.05 rejects the null hypothesis of equal variances. The normal distribution of data with equal variances was a prerequisite to use the *t*-test. When the normal distribution of data with no equality of variances was observed, the *t*-test with Cochran–Cox correction was performed. When one of the variables was of no normal distribution the non-parametric Mann–Whitney *U*-test was performed. Kruskal–Wallis one-way analysis of variance was performed to compare data in several groups, when at least data distribution in one group of no Gaussian distribution. The statistical significance was stated with post-hoc tests.

## 4. Results

The continuous gradual decline in cerebral blood flow volume with increasing age was observed in all study participants—the data are presented in Figure 1.

The correlation of the flow volume as sums of ICA, ECA and VA (*p* < 0.00001, *r* = −0.6231, *r*² = 0.3882) is presented in the Figure 1. The decline became more prominent with age. In the whole group, the annual decline in cerebral blood flow was 7.6 mL/year—Figure 1A. In the group aged below 65 years old, the annual decline in cerebral blood flow (CBF) was lower and estimated for 6.2 mL/year (*p* = 0.0019, *r* = −0.4052, *r*² = 0.1642)—Figure 1B. The decline became more prominent with age and in the group aged 65–75 reached 11.4 mL/year (*p* = 0.0121, *r* = −0.2759, *r*² = 0.0761)—Figure 1C. In the group aged above 75, it increased to 14.35 mL/year (*p* = 0.0074, *r* = −0.4222, *r*² = 0.1782)—Figure 1D.

Statistically significant volumetric differences were observed between the groups aged 65–69 years and 75–80 years (*p* = 0.0295), 65–69 years and > 80 years old (*p* < 0.000001), as well as between 70–74 years and > 80 years old (*p* = 0.00446)—Figure 2A. The gradual decline in total cerebral flow volume was mainly due to statistically significant flow volume decline in ICA (*p* = 0.00001, *r* = −0.3933, *r*² = 0.1547)—Figure 2B, and to lesser extent in ECA (*p* = 0.0011, *r* = −0.2916, *r*² = 0.0851)—Figure 2C, both for males and females. No statistically significant flow volume differences were observed between genders as well as between contralateral arteries in terms of ICA. Thus, the data are presented collectively, not disaggregated by gender. No statistically significant flow volume changes were observed in vertebral arteries (*p* > 0.07, *r* = −0.1636, *r*² = 0.0268)—Figure 2D.

No statistical significance in flow volume between contralateral arteries was observed, except vertebral arteries, left vertebral artery was dominant: 56 mL/min vs. 75 mL/min (*p* < 0.0001).

No male to female differences were observed between examined arteries except external carotid arteries, where flow volume was significantly higher in males (105.75 mL/min vs. 89 mL/min, *p* < 0.00007). As these observations have been previously described, they are not shown on the Figures.

The decrease in peak systolic velocity (PSV) (*p* = 0.002, *r* = −0.2392, *r*² = 0.0572), end-diastolic velocity (EDV) (*p* = < 0.00001, *r* = −0.4112, *r*² = 0.1691) in ICA as well as the decrease of PSV in ECA (*p* = 0.0017, *r* = −0.2612, *r*² = 0.0682) were observed. ICA EDV decrease was relatively more prominent than PSV decrease. No significant gender differences were observed in blood flow velocities thus the data are presented collectively in the Figure 3.

Values of flow volumes, velocities and proposed reference values with standard deviations among the study group are presented in the Table 3.

## 5. Discussion

Contemporarily Doppler ultrasound examination is a gold standard in the diagnostics of carotid stenosis and qualification to vascular intervention, enabling to asses both: the hemodynamics as well as other parameters connected with increased stroke risk [13]. The severity of stenosis, assessed mainly on the basis of changes in PSV, EDV, PSV ratio between CCA and stenosis, and lumen reduction, is the most important stroke risk factor. Rapid stenosis progression, large size of the plaque, low plaque echogenicity, increased juxta luminal black area, the presence of spontaneous embolization in transcranial doppler, plaque ulceration and impaired cerebrovascular reserve are the features increasing the risk of stroke in patients with asymptomatic carotid stenosis [7,14,15,16,17].

Cerebral blood flow, being the important, however difficult to assess quantity, is connected with cerebrovascular reserve. The estimation of total cerebral blood flow is mainly done with the use of radionuclide or techniques such as: positron emission tomography (PET), single-photon emission computed tomography (SPECT) and Xenon-enhanced CT scanning. Those methods allow to achieve reliable, reproducible and accurate results but due to their limited accessibility and high costs, are limited of use [9,18]. Despite the ongoing debate, there is no standard of cerebral blood flow volume, measured as a sum of volumes in the extracranial arteries [19]. The first description of sonographic assessment of cerebral blood flow (CBF) was first published by Schoening et. al. These authors also proved that the sonographic quantification of CBF is an accurate method, featured with high intradiane, interdiane and intraobserver and interobserver reproducibility and comparable with radionuclides methods [9,10,11,12].

The fact that total cerebral blood flow volume is subject to changes is known, however there is no abundant data in the literature concerning this problem, and the published research is conducted on relatively small study groups.

Schoening et al. sonographically examined the total cerebral blood flow (TCBF) changes in adolescents. According to their findings, the TCBF (a sum of flow volumes in ICA and VA), rapidly develops between 3 and 6.5 years old (raising from 687 ± 85 to 896 ± 110 mL/min), and consecutively decline from 6.5 years old to adulthood, reaching about 700 mL/min in 15 years old. The authors claim that the increase of TCBF was caused mainly by the statistically significant increase mainly in ICA flow volumes, and less prominently in VA. From 65 years on the marked decrease in flow volume in VA arteries was observed, without significant decrease of ICA flow volume. ECA flow volume was observed to continually increase with age. The authors did not observe sex-related differences [20].

In 1982, Umeatsu et al. measured the blood flow in common carotid artery using ultrasonic volume flowmeter (VFM). The authors stressed that the carotid blood flow varies with age. In subjects with carotid stenosis, the authors found decreased flow volume caused by atherosclerotic lesions, and elevated flow volume after endarterectomy. The authors also stressed that in the physiological conditions ICA 70%, comparing to external carotid flow, which is approximately that of vertebral flow (30%) [21].

Up to this day, several authors dealt with the problem of ultrasound examination of cerebral blood flow volume in healthy adults. Scheel et al. published in the Stroke their work in which the influence of age and sex on cerebral blood flow was examined. After examination of 78 healthy adults, the authors found a statistically significant inverse correlation between age and cerebral blood flow volume, caused mainly by the significant reduction in bilateral ICA flow volume, without significant changes in other arteries. The authors found that cerebral blood flow volume slightly decreases at a 3 mL/year rate [9]. Several other research studies using other imaging techniques like magnetic resonance imaging (MRI) confirmed this result, with the rate of CBF volume decrease ranging from 3.9 to 4.8 mL/year [9,22,23]. Phase contrast MRI imaging is thought to be a very accurate method of CBF assessment, however B-flow imaging gives very close, however significantly higher, results. Color Doppler and power Doppler techniques can overestimate the flow volume [18]. Therefore, determining the arterial lumen diameter is a factor of utmost importance to obtain reliable flow volume results, because even an 0.1 mm diameter difference may result in a few percent results over or underestimation.

Yazici et al. also demonstrated age-dependent cerebral blood flow volume decreases due to volumes and velocities decrease in CCA, ICA and VA. There were no sex differences in flow volume in extracranial arteries except ECA, in which the flow volume was significantly higher in men [19]. The results considering blood flow velocities are consistent with the ones obtained by Sheel et al., who observed volumetric flow only in the ICA [24]. The decrease of blood flow volume in the ICA in an ageing population, independent of gender, with no side-to-side differences was also proved by Schebesch et al. [25]. Albayrak et al. also demonstrated the decrease in peak systolic velocities, end diastolic velocities, and flow volumes in ICA and VA with increasing age [26]. Schöning et al., examining the group of healthy adults between 20 and 63 years old, did not noted the decrease in cerebral blood flow in this group. The authors observed decrease in flow velocities in ICA (PSV and EDV) and PSV in VA [10].

In this study, the decline in TCBF volume was noted, becoming more prominent with age. The annual decline in the group aged between 18–65 proved to be much smaller than in individuals over 65 years old. The decrease is the result of significant decrease of flow volume mainly in ICA and, to lesser extent, ECA. The decrease in ICA, PSV, EDV, and ECA PSV is convergent with obtained volumetric results, being in accordance with results published by other authors. However, it is worth stressing that the cerebral blood flow volume decline is more rapid in elderly people than in younger aged groups, accelerating rapidly after 70–75 years old, which was not published previously. These data are in accordance with the contemporary knowledge concerning cerebral blood flow and physiological brain aging processes. The brain perfusion and its weight changes throughout life. Its mass rapidly increases by the age of 6, reaching about 92% of its maximum weight, which is obtained in about 18 years old adults [20,27,28,29]. Those changes are accompanied by perfusion changes. In healthy adults, brain perfusion is estimated to 50 mL/100g/min, with higher flow in grey matter of 80 mL/100g/min and lower in white matter of 20 mL/100g/min [30,31]. It is proven that, after exceeding the age of forty, the volume and mass of brain begin to decline: initially at the rate of 5% per year, increasing over 70 years old [30,31,32]. In the ageing brain, white matter lesions (WML) are observed. Common also in asymptomatic patients, they are connected with increased cardiovascular risk, reduction of cerebral reactivity, blood flow and vascular density [32,33,34,35]. Elevated blood pressure also influences the brain aging processes, leading to faster brain atrophy, especially accentuated in grey matter [32,36]. This factor also stresses the importance of excluding the patients with elevated blood pressure from the examination.

Throughout the years, there were also no consensus concerning the reference vertebral arteries blood flow volumes. In the 1980s, with vertebral flow below 200 mL/min, vertebrobasilar insufficiency was suspected [37], however, later studies reported values between 100 mL/min and 300 mL/min in healthy individuals [19,22,23,24]. Generally, up to three quarters of patients have one dominant vertebral artery, and this phenomenon mainly occurs at left side [9,10,22,23,24]. The lowest vertebral volume values in this study oscillated about 70 mL/min with the highest values exceeding 200 mL/min. All the patients were asymptomatic with good waveform spectrum in vertebral arteries, which confirms that flow volume in VA below 100 mL/min may not cause vertebrobasilar insufficiency. In our study, the left vertebral artery was dominant in the majority of patients, which is consistent with previously published studies.

Minor discrepancies in the volumetric measurements might originate from the flow volume calculation method. The volume of blood is a product of cross-sectional area multiplied by time-averaged velocity (TAV), which may be estimated either from the maximum frequency TAMAX (also known as TAP—time-averaged peak velocity) or from TAMEAN (intensity weighted mean frequency) [38,39]. Volume values calculated with time-averaged peak frequency are featured with overestimating tendency [38,39,40]. When the flow pattern in the examined vessel is “flat”—flow velocities in peripheral parts of the vessel are almost the same as in the central part, both TAMAX and TAMEAN have similar values. In carotid arteries, where the flow pattern is considered parabolic, TAMEAN may have slightly lower values as TAMAX [38,39,40]. TAMEAN values depends also on sample volume—the gate should be broad enough to measure central and peripheral bloodstream. Too small sample volume might cause overestimation of TAMEAN, representing only blood layers in the measured part of the vessel [38,39,40]. In physiological conditions in carotid arteries, the major part of the flow occurs during systole, with the flow pattern resembling a flat pattern. In end-diastole, flow becomes parabolic [26,38,39,40]. Measurement of the arterial lumen and proper sample volume positioning are of key importance in achieving reliable measurements. In this study sample, volume covered 0.5 to 2/3 of arterial lumen to provide reliable TAMEAN estimation. The flow in the ICA was measured 3–4 cm above the carotid bifurcation where it regains laminar character, avoiding false results due to changes in velocity in the vicinity of carotid bulb. The ECA flow was measured distally to the origin of superior thyroid artery. In euthyroid patients, thyroid blood flow may exceed 100 mL/min (it is estimated as 5 mL/min/g) and superior thyroid artery is responsible for a considerable part of it [41].

Cerebral blood flow is estimated by the majority of authors as a sum of the flow in ICA and VA. In physiological conditions, ECA provides very little blood supply to the central nervous system. However, in case of the presence of significant stenosis of the ICA, ECA becomes the vital collateral blood supply pathway to the brain structures, which in Doppler ultrasound examination, might be featured with decrease in flow resistance, increase in end diastolic velocity and flow volume. In order to be able to assess the degree of compensation, the reference values of cerebral blood flow volume, including the ECA, should be known.

## 6. Conclusions

This study provides a new insight into physiological changes of carotid hemodynamics: cerebral blood flow volume reference values in patients beyond 65 years were determined, gradual decline in cerebral blood flow, caused mainly by the significant reduction of ICA flow volume and to lesser extent ECA flow volume, was observed in healthy volunteers. The significant decline in the peak systolic velocity (PSV) and end diastolic velocity (EDV) in the ICA, and PSV in the ECA, was observed.

The knowledge of processes concerning changes in cerebral and carotid blood flow in a healthy population may be a source of important additional information in diagnostics of patients with carotid stenosis.

## Figures and Tables

**Figure 1 jcm-09-01375-f001:**
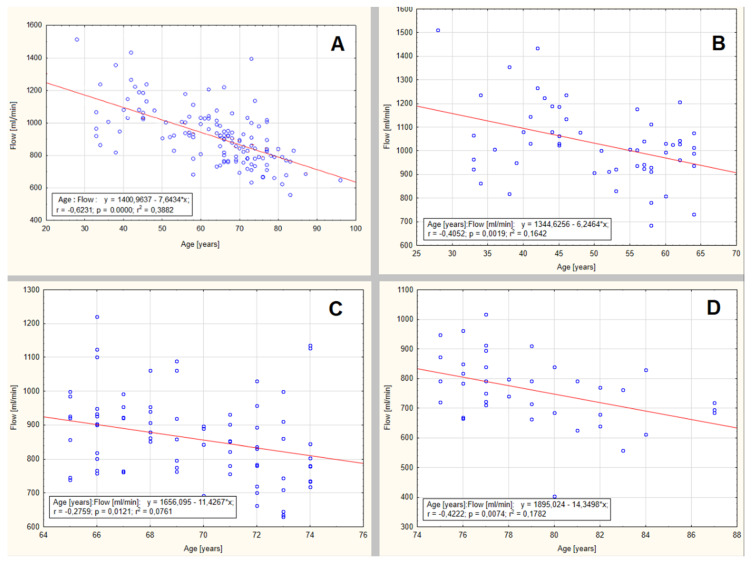
Regression analysis of cerebral blood flow volume—sum of flow volumes in the internal carotid artery (ICA), external carotid (ECA) and vertebral artery (VA). (**A**) Statistically significant decrease of 7.6 mL/min in whole study group (*p* < 0.00001, *r* = −0.6231, r² = 0.3882). (**B**) Smaller decline of 6.2 mL/year in the group aged below 65 years (*p* = 0.0019, *r* = −0.4052, *r*² = 0.1642). (**C**) More prominent with age and in the group aged 65–75 reaching 11.4 mL/year (*p* = 0.0121, *r* = −0.2759, *r*² = 0.0761). (**D**) Increase to 14.35 mL/year above 75 years old. (*p* = 0.0074, *r* = −0.4222, *r*² = 0.1782).

**Figure 2 jcm-09-01375-f002:**
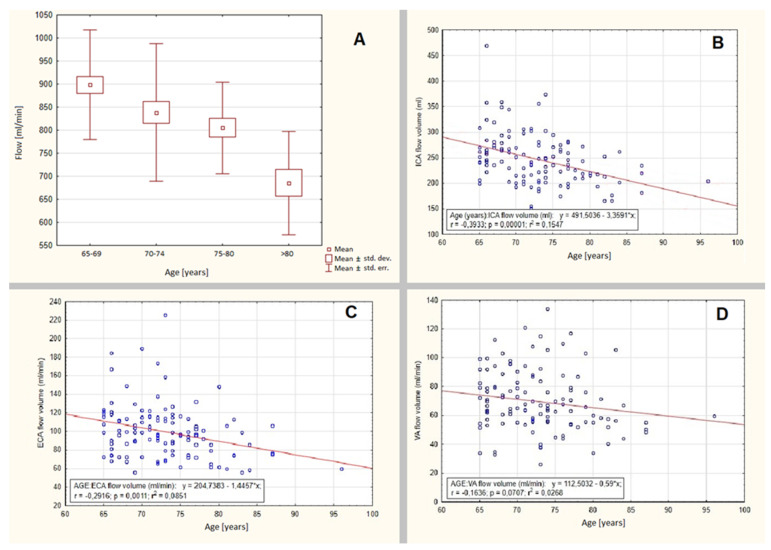
The gradual decline of cerebral flow volume in patients over 65 years old. (**A**) Statistically significant volumetric differences between the groups aged: 65–69 and 75–80 years (*p* = 0.0295), 65–69 and >80 years old (*p* < 0.000001), 70–74 and >80 years old (*p* = 0.00446). (**B**) Statistically significant ICA flow volume decline (*p* = 0.00001, *r* = −0.3933, *r*² = 0.1547). (**C**) Less prominent ECA flow volume decline (*p* = 0.0011, *r* = −0.2916, *r*² = 0.0851). (**D**) No statistically significant flow volume changes in vertebral arteries with age (*p* > 0.07).

**Figure 3 jcm-09-01375-f003:**
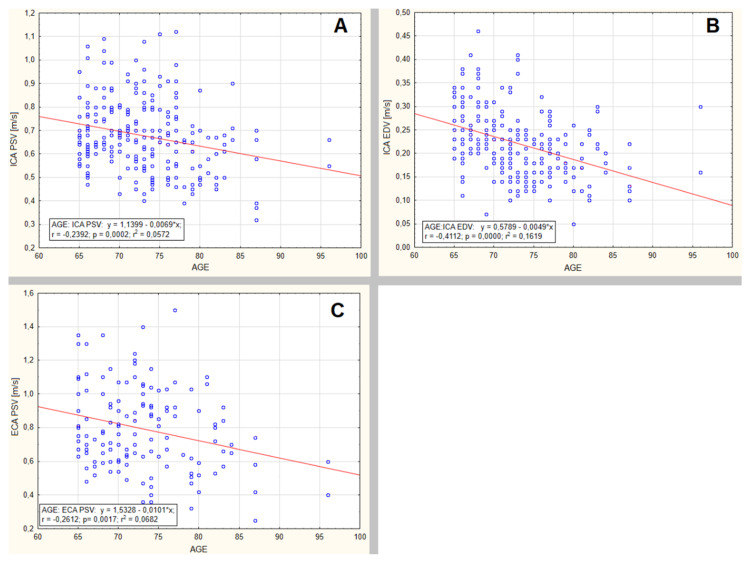
Flow velocity changes in internal and external carotid arteries. (**A**) The significant decrease of peak systolic velocity (PSV) (*p* = 0.002, *r* = −0.2392, *r*² = 0.0572). (**B**) The significant decrease of end-diastolic velocity (EDV) (*p* = <0.00001, *r* = −0.4112, *r*² = 0.1691) in ICA. ICA EDV decrease is relatively more prominent. (**C**) The significant decrease of PSV in ECA (*p* = 0.0017, *r* = −0.2612, *r*² = 0.0682).

**Table 1 jcm-09-01375-t001:** Group characteristics.

Age Group	Number of Patients	Average Age (Years)	Number of Females	Average Age of Females (Years)	Number of Males	Average Age of Males (Years)	Age Median in Females (Years)	Age Median in Males (Years)	Age Standard Deviation in Females (Years)	Age Standard Deviation in Males (Years)
65–69 years	42	66.9	21	67	21	66.8	67	66	1.5	1.4
70–74 years	41	72.2	16	72.1	25	72.3	72	72	1.2	1.5
75–79 years	24	76.8	11	76.8	13	76.8	77	77	1.0	1.6
≥80 years	16	83.7	13	83.5	3	84.7	82	87	4.21	4.0
<65 years	56	50.3	34	51.2	22	49	54	45	10.6	10.1

**Table 2 jcm-09-01375-t002:** Inclusion and exclusion criteria.

**Inclusion Criteria–Study Group**	
1	Age ≥ 65 years
2	Informed consent before the examination
3	No hemodynamically significant carotid atherosclerotic lesions, causing blood flow disturbances (ICA stenosis < 30%)
4	No exclusion criteria
**Exclusion Criteria–Study Group**	
1	Age < 65 years
2	No informed consent given before the examination
3	Internal Carotid Artery stenosis > 30%
4	Stenosis of Common Carotid, External Carotid or Vertebral Artery
5	Concomitant diseases: uncontrolled hypertension, ischemic heart disease, heart insufficiency, positive history of heart infraction, positive history of stent implantation to coronary or any other arteries, cardiac arrhythmia, tachycardia, bradycardia, congenital vascular or heart failure, positive history of vascular interventions, presence of endocrine diseases: thyroid goiter, hyper-, hypothyroidism diabetes, adrenal diseases, positive history of thyroid surgery, smoking, alcohol use.
6	Positive history of ischemic stroke, TIA symptoms or other neurological symptoms.

TIA: transient ischemic attack; ICA: internal carotid artery.

**Table 3 jcm-09-01375-t003:** Flow volume values and velocities in study group.

Group-Age	65–69	70–74	75–80	>80
Mean (mL/min)	898.5	838.5	805.1	685.7
Std. err (mL/min)	18.4	23.3	20.3	29.0
Std. dev (mL/min)	119.1	148.9	99.3	112.3
Proposed reference value (mL/min)	898.5 ± 119.1	838.5 ± 148.9	805.1 ± 99.3	685.7 ± 112.3
ICA volume (mL/min)	273.8 ± 60.5	237.9 ± 54.3	240.1 ± 47.3	203.3 ± 42.7
VA volume (mL/min)	71.8 ± 32.3	70.3 ± 28.2	60.5 ± 25	57.3 ± 18.5
ECA volume (mL/min)	103.6 ± 32.9	104.2 ± 32.7	91.5 ± 23	81 ± 35
ICA PSV (m/s)	0.72 ± 0.14	0.67 ± 0.15	0.68 ± 0.17	0.59 ± 0.14
ICA EDV (m/s)	0.26 ± 0.07	0.21 ± 0.06	0.20 ± 0.05	0.18 ± 0.06
VA PSV (m/s)	0.45 ± 0.11	0.45 ± 0.14	0.44 ± 0.12	0.41 ± 0.11
VA EDV (m/s)	0.13 ± 0.06	0.12 ± 0.05	0.13 ± 0.04	0.13 ± 0.04
ECA PSV (m/s)	0.8 ± 0.24	0.81 ± 0.25	0.78 ± 0.26	0.67 ± 0.21
ECA EDV (m/s)	0.1 ± 0.05	0.13 ± 0.07	0.13 ± 0.05	0.12 ± 0.05

ICA, internal artery; ECA, external carotid; VA, vertebral artery.

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
