# Peer review of "Volumetric Carotid Flow Characteristics in Doppler Ultrasonography in Healthy Population Over 65 Years Old"

_jcm, 2020, doi:10.3390/jcm9051375_

Round 1
Reviewer 1 Report
This is a well conducted and well written paper.
Methods are stringent and results are of interest.
I suggest to delete the word "invasive" at line 177 when reporting radionuclide techniques.
References need to be corrected as per journal style.
Author Response
Warsaw, 4 May 2020
Piotr Kaszczewski Ph.D. M.D.
Department of General, Endocrine and Vascular Surgery
Medical University of Warsaw, Poland
Banacha 1a, 02-097 Warsaw
Dir Sir or Madam,
On behalf of all authors I would like to thank You for the time and effort devoted to prepare the review as well as for the constructive comments, which provided valuable insights, indispensable in refining manuscript content.
I would like to ensure Your Party that all issues were thoroughly corrected, according to the review. I allow myself to present the detailed list of revisions in manuscript body.
The list of changes in the manuscript body:
- Abstract: Background part reformatted: divided into more sentences. Sentences changed to be more precise. Detailed changes include:
- Line 10-11: word “known” changed to “established” and “with age being a factor influencing measurements.” added as the second part of the introducing sentence.
- Line 11-12: a part of the sentence “…,but there are no volumetric standards of the flow in extracranial arteries.” reformatted as the new sentence: “However, there are no volumetric standards for the flow in extracranial arteries”. A words “Study aim covered” changed to “The aim of the study was related to…”
- Line 13: word “old” added after “>65 years”.
- Abstract: Methods part reformatted. Detailed changes include:
- Line 13: word “assessment” changed to “measurements”
- Line 14: word “was” changed to “were”
- Line 15: word “old” added after “>65 years” and “<65 years”
- Abstract: Results part reformatted, new sentences formed. Detailed changes include:
- Line 16: word “stated” changed to “observed”
- Line 19: “1 and 3,4, and 3 and 4” changed to “1 and 3,4, as well as 3 and 4”.
- Line 20: “decreases increasingly” changed to “decreases gradually”
- Line 21-22: “as a result of ICA (p=0,00001) and to lesser extent ECA (p=0,0011) flow volume decline” reformatted as a new sentence: “This is a result of flow volume decline in ICA (p=0,00001), and to lesser extent, ECA (p=0,0011)”.
- Abstract: Conclusion part reformatted, new sentence added to make the whole abstract more comprehensible with current literature. Detailed changes include:
- Line 25: word ”pose” changed to “play”
- Line 25-27: a new sentence added “Doppler assessment of cerebral flow volume may create an interesting tool for identifying patients with diminished cerebrovascular reserve and higher risk of ischaemic symptoms occurrence.”
- Line 41: word “the” added before “common carotid artery…”
- Line 53: word “in” added before “the diagnosis and monitoring…”
- Line 59: spelling mistake corrected “in”
- Lines 122-137: References to all figures are changed to be mentioned in a systematic manner, i.e.: “… - Figure 1A”. Section divided into shorter sentences. Detailed changes include:
- Line 122: word “presenting” changed to word “of”
- Line 123: word “figure 1A” changed to “Figure 1”, word “becames” changed to became”
- Line 124: word “(Figure 1A)” changed to “- Figure 1A”
- Line 127: word “reaches” changed to “reached”. A sentence devided into two smaller sentences: “Figure 1C, while in the group” changed to “– Figure 1C. In the group”
- Lines 129-131 – sentence “The differences in flow volume among are presented on the Figure 2 A” changed to “ – Figure 2A”.
- Lines 134-136: sentence divided into two shorter sentences: “arteries in terms of ICA, thus the data are presented” changed to “arteries in terms of ICA. Thus the data are presented…”
- Line 136: “no disaggregated by gender” changed to “not disaggregated by gender”
- Figure 2: Caption changed in order to be coherent with Figure 1 caption. Detailed changes include:
- Line 140: word “Figure 2A – “ added before “statistically significant differences…”
- Line 141: word “volumetric” added before “differences…”, words “were observed” removed, “:” added after “groups aged”
- Line 142-143: word “Figure 2B –“ added before “statistically significant ICA flow volume differences…”, “the differences were caused by” removed.
- Line 143-144: words “Figure 2C – less prominent ECA flow…” added instead of “and to lesser extent ECA flow”
- Lines 144-145: words “Figure 2D –“ moved at the beginning of the sentence
- Figure 3: Caption changed in order to be coherent with Figure 1 and corrected Figure 2 captions. Detailed changes include:
- Line 160: “Flow velocity changes in internal and external carotid arteries.” added after “Figure 3.”. Word “Figure 3A –“ added befeore “ the significant…”.
- Line 161: “.” used after “(… r2=0,0572)”. “Figure 3B –“ added before “the significant decrease”
- Line 163: “Figure 3C –“ added before “the significant decrease…” and moved down to a line 163.
- Line 173: “:” erased.
- Line 180: word "invasive" is removed
- Line 181: word “and” added before “Xenon-enhanced…”.
- Line 193: word “founding’s” changed to “findings”.
- Line 266: “featured” spelling corrected
- Line 272: “blood lawyers” change to “blood layers”.
- Line 280: “superior thyroid artery” instead of “external carotid artery”.
- Line 283: words “Last but not least” erased.
- Line 286: Abbreviation “CDD” changed to “Doppler ultrasound”.
- References: All references corrected to a required format:
- Author 1, A.B.; Author 2, C.D. Title of the article. Abbreviated Journal Name Year, Volume, page range.
We trust You will find the revised version of manuscript suitable for publication in Journal of Clinical Medicine.
Yours faithfully,
Piotr Kaszczewski

Reviewer 2 Report
This is a well presented study that in my opinion need few grammatical og conceptual changes to be ready for publication.
Major issue.
The abstract needs to be restructured completely to make the results more comprehensible and to put the results in a relevant context with the current litterature.
Minor issues.
Abstract line 2: Add of the after assessment and restructure sentence.
Divide abstract into more sentences and exclude symbols like - from abstract.
Line 37: add the before common.
Line 49: add in before first the.
Line 118-134: Please use less symbols like - and more full stops. Please divide section into more shorter and precise sentences. Please mention the figures in a systematic manner.
Figure 2: Please change caption. Start sentence by A: or (A) and so forth.
Figure 2: Please change caption. Start sentence by A: or (A) and so forth.
Line 170: Erase :
Line 179: Add and before Xenon.
Line 190: Change to findings.
Line 169: Blood layers instead.
Line 277: Thyroid artery instead of what was written.
Line 280: Consider erasing Last but not least... seems unnecessary.
Line 283: Please do not use CDD abbreviation when it has not been introduced.
Author Response
Warsaw, 4 May 2020
Piotr Kaszczewski Ph.D. M.D.
Department of General, Endocrine and Vascular Surgery
Medical University of Warsaw, Poland
Banacha 1a, 02-097 Warsaw
Dir Sir or Madam,
On behalf of all authors I would like to thank You for the time and effort devoted to prepare the review as well as for the constructive comments, which provided valuable insights, indispensable in refining manuscript content.
Having received the review, we put all our efforts to implement necessary changes and prepare the manuscript body in accordance with Your suggestions. We hope You will find the improved version of abstract presenting results in more comprehensible manner and with connections with current literature. I would like to ensure Your Party that all minor issues were thoroughly corrected, according to the review. I allow myself to present the detailed list of all revisions.
The list of changes in the manuscript body:
- Abstract: Background part reformatted: divided into more sentences. Sentences changed to be more precise. Detailed changes include:
- Line 10-11: word “known” changed to “established” and “with age being a factor influencing measurements.” added as the second part of the introducing sentence.
- Line 11-12: a part of the sentence “…,but there are no volumetric standards of the flow in extracranial arteries.” reformatted as the new sentence: “However, there are no volumetric standards for the flow in extracranial arteries”. A words “Study aim covered” changed to “The aim of the study was related to…”
- Line 13: word “old” added after “>65 years”.
- Abstract: Methods part reformatted. Detailed changes include:
- Line 13: word “assessment” changed to “measurements”
- Line 14: word “was” changed to “were”
- Line 15: word “old” added after “>65 years” and “<65 years”
- Abstract: Results part reformatted, new sentences formed. Detailed changes include:
- Line 16: word “stated” changed to “observed”
- Line 19: “1 and 3,4, and 3 and 4” changed to “1 and 3,4, as well as 3 and 4”.
- Line 20: “decreases increasingly” changed to “decreases gradually”
- Line 21-22: “as a result of ICA (p=0,00001) and to lesser extent ECA (p=0,0011) flow volume decline” reformatted as a new sentence: “This is a result of flow volume decline in ICA (p=0,00001), and to lesser extent, ECA (p=0,0011)”.
- Abstract: Conclusion part reformatted, new sentence added to make the whole abstract more comprehensible with current literature. Detailed changes include:
- Line 25: word ”pose” changed to “play”
- Line 25-27: a new sentence added “Doppler assessment of cerebral flow volume may create an interesting tool for identifying patients with diminished cerebrovascular reserve and higher risk of ischaemic symptoms occurrence.”
- Line 41: word “the” added before “common carotid artery…”
- Line 53: word “in” added before “the diagnosis and monitoring…”
- Line 59: spelling mistake corrected “in”
- Lines 122-137: References to all figures are changed to be mentioned in a systematic manner, i.e.: “… - Figure 1A”. Section divided into shorter sentences. Detailed changes include:
- Line 122: word “presenting” changed to word “of”
- Line 123: word “figure 1A” changed to “Figure 1”, word “becames” changed to became”
- Line 124: word “(Figure 1A)” changed to “- Figure 1A”
- Line 127: word “reaches” changed to “reached”. A sentence devided into two smaller sentences: “Figure 1C, while in the group” changed to “– Figure 1C. In the group”
- Lines 129-131 – sentence “The differences in flow volume among are presented on the Figure 2 A” changed to “ – Figure 2A”.
- Lines 134-136: sentence divided into two shorter sentences: “arteries in terms of ICA, thus the data are presented” changed to “arteries in terms of ICA. Thus the data are presented…”
- Line 136: “no disaggregated by gender” changed to “not disaggregated by gender”
- Figure 2: Caption changed in order to be coherent with Figure 1 caption. Detailed changes include:
- Line 140: word “Figure 2A – “ added before “statistically significant differences…”
- Line 141: word “volumetric” added before “differences…”, words “were observed” removed, “:” added after “groups aged”
- Line 142-143: word “Figure 2B –“ added before “statistically significant ICA flow volume differences…”, “the differences were caused by” removed.
- Line 143-144: words “Figure 2C – less prominent ECA flow…” added instead of “and to lesser extent ECA flow”
- Lines 144-145: words “Figure 2D –“ moved at the beginning of the sentence
- Figure 3: Caption changed in order to be coherent with Figure 1 and corrected Figure 2 captions. Detailed changes include:
- Line 160: “Flow velocity changes in internal and external carotid arteries.” added after “Figure 3.”. Word “Figure 3A –“ added befeore “ the significant…”.
- Line 161: “.” used after “(… r2=0,0572)”. “Figure 3B –“ added before “the significant decrease”
- Line 163: “Figure 3C –“ added before “the significant decrease…” and moved down to a line 163.
- Line 173: “:” erased.
- Line 180: word "invasive" is removed
- Line 181: word “and” added before “Xenon-enhanced…”.
- Line 193: word “founding’s” changed to “findings”.
- Line 266: “featured” spelling corrected
- Line 272: “blood lawyers” change to “blood layers”.
- Line 280: “superior thyroid artery” instead of “external carotid artery”.
- Line 283: words “Last but not least” erased.
- Line 286: Abbreviation “CDD” changed to “Doppler ultrasound”.
- References: All references corrected to a required format:
- Author 1, A.B.; Author 2, C.D. Title of the article. Abbreviated Journal NameYear, Volume, page range.
We trust You will find the revised version of manuscript suitable for publication in Journal of Clinical Medicine.
Yours faithfully,
Piotr Kaszczewski
